# Influencing Factors Identification and Prediction of Noise Annoyance—A Case Study on Substation Noise

**DOI:** 10.3390/ijerph19148394

**Published:** 2022-07-09

**Authors:** Guoqing Di, Yihang Wang, Yao Yao, Jiangang Ma, Jian Wu

**Affiliations:** 1College of Environmental and Resource Sciences, Zhejiang University, Hangzhou 310058, China; 22114087@zju.edu.cn (Y.W.); 21914030@zju.edu.cn (Y.Y.); 2State Grid Shaanxi Electric Power Research Institute, Xi’an 710054, China; majg1988@mail.nwpu.edu.cn (J.M.); grid_epkl@163.com (J.W.)

**Keywords:** noise annoyance, non-acoustic factors, influence weight, prediction model, substation noise

## Abstract

Noise-induced annoyance is one person’s individual adverse reaction to noise. Noise annoyance is an important basis for determining the acceptability of environmental noise exposure and for formulating environmental noise standards. It is influenced by both acoustic and non-acoustic factors. To identify non-acoustic factors significantly influencing noise annoyance, 40 noise samples with a loudness level of 60–90 phon from 500–1000 kV substations were selected in this study. A total of 246 subjects were recruited randomly. Using the assessment scale of noise annoyance specified by ISO 15666-2021, listening tests were conducted. Meanwhile, basic information and noise sensitivity of each subject were obtained through a questionnaire and the Weinstein’s noise sensitivity scale. Based on the five non-acoustic indices which were identified in this study and had a significant influence on noise annoyance, a prediction model of annoyance from substation noise was proposed by a stepwise regression. Results showed that the influence weight of acoustic indices in the model accounted for 80% in which the equivalent continuous A-weighted sound pressure level and the sound pressure level above 1/1 octave band of 125 Hz were 65% and 15%, respectively. The influence weight of non-acoustic indices entering the model was 20% in which age, education level, noise sensitivity, income, and noisy degree in the workplace were 8%, 2%, 4%, 4%, and 2%, respectively. The result of this study can provide a basis for factors identification and prediction of noise annoyance.

## 1. Introduction

Noise-induced annoyance is one person’s dissatisfaction, bother, annoyance, disturbance, and other adverse reactions to noise. Noise annoyance is influenced by both acoustic and non-acoustic factors [1,2,3]. Acoustic factors can be divided into psychoacoustic factors and non-psychoacoustic factors. Previous studies indicated that psychoacoustic factors included loudness (N), fluctuation strength (F), roughness (R), sharpness (S), and tonality (T) [4,5]. Non-psychoacoustic factors included sound pressure level (SPL), equivalent continuous A-weighted sound pressure level ($L_{\mathrm{Aeq}}$), equivalent continuous C-weighted sound pressure level ($L_{\mathrm{Ceq}}$), the difference between C-weighted sound pressure level and A-weighted sound pressure level ($L_{C-A}$), octave band sound pressure level ($L_{31.5\mathrm{Hz}}$, $L_{63\mathrm{Hz}}$, $L_{125\mathrm{Hz}}$, $L_{250\mathrm{Hz}}$, $L_{500\mathrm{Hz}}$, $L_{1\mathrm{kHz}}$, $L_{2\mathrm{kHz}}$, $L_{4\mathrm{kHz}}$, $L_{8\mathrm{kHz}}$, $L_{16\mathrm{kHz}}$), and the peak of sound pressure levels above all octave bands ($L_{\max}$) [6,7]. The main acoustic factors influencing noise annoyance are different for different noise sources due to their different acoustic characteristics. Alayrac et al. [6] divided different industrial noise sources into six types and distinguished the main acoustic factors influencing noise annoyance according to their acoustic characteristics. Non-acoustic factors can be divided into personal factors and socio-economic factors [2]. Previous studies showed that personal factors included age, education level, noise sensitivity, etc. [1,2]. Socio-economic factors included occupation and attitudes towards noise sources, etc. [1,2]. To improve the accuracy of prediction models of noise annoyance, it is necessary to identify non-acoustic indices which have a significant influence on noise annoyance.

Most of the existing prediction models of noise annoyance only considered the influence of acoustic indices [6,7,8]. Noise-induced annoyance cannot be explained only by acoustic indices as non-acoustic indices have a similar importance [9,10]. The prediction models only containing acoustic indices can predict the average noise annoyance of the public and cannot reflect the difference of individual noise annoyance. The prediction models of noise annoyance including acoustic and non-acoustic indices can recognize the difference of individual noise annoyance and improve the accuracy of models. For example, the prediction model of annoyance from road traffic noise developed by Gille et al. [11] considered the influence of subjects’ noise sensitivity. The prediction models of annoyance from road traffic and aircraft noise developed by Preisendörfer et al. [12] considered the influence of subjects’ environmental concern and income. However, non-acoustic indices significantly influencing noise annoyance were not identified systematically in the process of developing the models above [13].

In this study, substation noise was chosen as a case study and the subjects’ noise annoyance to each noise sample was obtained through listening tests. According to the range of a single non-acoustic index, it was divided into several subintervals. Subjects were divided into several subgroups according to the subintervals. A paired samples *t*-test was used to examine whether the annoyance of noise samples had a significantly difference between two subgroups of subjects. That is, non-acoustic indices significantly influencing noise annoyance were identified according to the paired samples *t*-test results. On this basis, a prediction model of noise annoyance including acoustic and non-acoustic indices was developed by a stepwise regression. The influence weight of each index was explored according to the model.

## 2. Materials and Methods

### 2.1. Subject Recruitment and Information Collection

A total of 246 volunteers with self-reported normal hearing were randomly recruited as experimental subjects. Age, gender, education level, income, and noisy degree in the workplace of each subject were obtained through a questionnaire. Noise sensitivity of each subject was obtained through the Weinstein’s noise sensitivity scale [14]. Noise sensitivity belonged to personal privacy, so only 138 subjects were willing to participate in the survey. The noise sensitivity data of 124 subjects were collected after questionnaires with missing information, not normatively filled out in accordance with requirements, or having obviously inaccurate information were rejected. A data collection fee was paid for each subject in the listening tests.

### 2.2. Noise Samples

The main noise sources in substations included transformers, reactors, and capacitors. A total of 40 noise samples from 500–1000 kV substations were recorded at different horizontal distances away from the main noise sources of substations by Artificial Head HMS IV.1 (Head Acoustics, Herzogenrath, Germany). The height of sampling points was 1.5 m above the ground. The loudness level of noise samples was 60–90 phon. The equivalent continuous A-weighted sound pressure level of noise samples was 30–75 dBA. Scarcely being disturbed by other noise, each noise sample with a duration of 5 s was extracted from the primitive sample by ArtemiS 10.0. A total of 40 noise samples were randomly sorted 3 times and an interval of 7 s (5 s evaluation time + 1s warning beep of starting + 1 s mute) was inserted between adjacent samples to form a sequence of noise samples used in listening tests. Figure 1 and Figure 2 showed the 1/1 octave band spectra and time-frequency signal of two typical noise samples from 500 kV and 1000 kV substations, respectively. The A-weighted sound pressure level of two typical noise samples are 66.5 dBA and 60.9 dBA, respectively. As shown in Figure 1 and Figure 2, low-frequency components are dominant in these two substations’ noise and high-frequency sound energy is more presented in the noise sample from the 1000 kV substation than that from the 500 kV substation.

### 2.3. Room and Equipment for Listening Tests

Listening tests were conducted in a soundproof room (3 m × 2 m × 2 m) with background noise below 25 dBA. The experimental audio playback system consisted of a computer installing ArtemiS 10.0, a digital equalizer (PEQ V, Head acoustics, Herzogenrath, Germany), a headphone signal distribution amplifier (HAD IV.1, Head acoustics, Herzogenrath, Germany), and four high-quality headphones (HD600, Sennheiser, Wedemark, Germany). Compared with actual sound, the sound reproduced by the playback system had an uncertainty [15]. The playback system needs to be calibrated to ensure that the reproduced sound was consistent with reality. The equipment used in this study passed a rigorous test by the instrument supplier before they were delivered. Before the listening tests, the headphones of the playback system would be put on the Artificial Head HMS IV.1 to further check the consistency of the reproduced sound and actual sound. The binaural sound signal collected by Artificial Head HMS IV.1 were compared with that reproduced by the playback system to check the consistency of these two signals. The results of the test showed that the playback system did not have a distinct distortion from 20 Hz to 20,000 Hz and would not affect the reproduction of low-frequency components of noise samples.

### 2.4. Experimental Procedures

Before the listening tests, the subjects listened to experimental instructions and signed an informed consent. The content of the informed consent includes the purpose of this study, the experimental procedures, the risks, and the compensation. After a period intended for calming down, the subjects accepted noise exposure through headphones. A preliminary listen was performed and subjects were asked to listen to the sound samples with the largest and smallest loudness in the experimental sound samples group before a formal experiment, which can induce subjects to establish a stable psychological scale in listening tests. Thus, the accuracy of the annoyance rating from subjects can be improved. An 11-point numerical rating scale in which 0 represented not annoyed at all and 10 represented extremely annoyed specified by ISO 15666-2021 was used in the listening tests [16]. After each noise sample was played, subjects chose a number from 0 to 10 to evaluate the annoyance rating of the sample.

### 2.5. Statistical Analysis

If the difference between any two evaluation results for a same noise sample from a same subject is greater than 2, the noise sample would be regarded as a misjudged noise sample from this subject and the data of misjudged noise samples will be removed in data analysis [17]. If the ratio of the amount of misjudged noise samples to the total amount of noise samples is higher than 30%, this subject would be regarded as an invalid subject and all data from the subject would be eliminated. The removal ratios of misjudged noise samples and invalid subjects were 22.6% and 24.4%, respectively.

In listening tests, the subject acted as a “measuring instrument” for noise annoyance. Noise annoyance of a same noise sample from different subjects or noise annoyance of a same noise sample in different experimental groups from a same subject may have a difference due to the different psychological scales of the subjects. Besides, noise annoyance of a same noise sample could be different due to a different HRTF in different subjects [18]. In this study, the difference of HRTF among different subjects was not considered. A certain noise sample was not adjusted according to HRTF of each subject, so it could not ensure that the loudness of the noise sample heard by each subject was the same. Therefore, it was necessary to calibrate the measuring results from the “measuring instrument” in order to ensure that the measuring results were based on a same scale. In order to reduce the difference of noise annoyance caused by individual factors, such as HRTF of subjects, a master scaling method [17] was adopted to calibrate noise annoyance data of a same noise sample from different individuals after the misjudged data were eliminated in this study.

After data calibration, the mean noise annoyance ($A_{m}$) of each noise sample is calculated according to Equation (1)
(1)$$A_{m}=\frac{1}{V}\overset{V}{\underset{i=1}{\sum }}A_{i}$$
where $A_{i}$ is the annoyance rating from the *i*th subject in the numerical rating scale and *V* is the number of valid subjects.

According to the range of a single non-acoustic index, it was divided into several subintervals. Subjects were divided into several subgroups according to the subintervals. In this study, 40 noise samples were used in the listening tests. The $A_{m}$ of a same noise sample was different in different subgroups. Calculating and analyzing the difference of $A_{m}$ of each noise sample between two subgroups of subjects, a paired samples *t*-test was used to examine whether the annoyance of 40 noise samples had a significant difference between two subgroups of subjects. If there was not a significant difference between the noise annoyance of two adjacent subgroups, these two subgroups would be merged to one group; otherwise, these two subgroups would not be combined. The steps above were repeated until all adjacent subgroups cannot be merged further. Finally, if the noise annoyance of any two adjacent subgroups in all subgroups all have a significant difference after the combination was completed, the non-acoustic index would be regarded as an index which could significantly influence noise annoyance.

The values of 20 acoustic indices (mentioned in the Introduction) of 40 experiment noise samples were calculated by ArtemiS10.00. The correlation between $A_{m}$ and each acoustic index was analyzed by a linear regression and the determination coefficient was calculated. According to the acoustic index value (e.g., *N*) and $A_{m}$ of 40 noise samples, the Pearson correlation coefficient between each acoustic index and $A_{m}$ was calculated to examine whether these two variables were significantly linearly correlated.

Based on the acoustic indices being significantly linearly correlated with noise annoyance and the non-acoustic indices significantly influencing noise annoyance, a prediction model of noise annoyance was developed by a stepwise regression. According to the standard regression coefficient of each index in the model, the influence weight of each index on noise annoyance was explored.

SPSS20.0 was used for statistical analysis of data in this study.

## 3. Results and Discussion

### 3.1. Subjects’ Information

The subjects’ basic information was shown in Table 1. The range of each non-acoustic index such as age, gender, education level, noise sensitivity, income, and noisy degree in the workplace of the subjects was divided into several subintervals. Subjects were divided into several subgroups according to the subintervals of each non-acoustic index. The initial subgroups were shown in Table 1. As shown in Table 1, subjects were divided into six subgroups (a1–a6) according to their age, two subgroups (b1–b2) according to their gender, five subgroups (c1–c5) according to their education level, two subgroups (d1–d2) according to their noise sensitivity, four subgroups (e1–e4) according to their income, and four subgroups (f1–f4) according to noisy degree in their workplace.

The results of the paired samples *t*-test between the noise annoyance of any two adjacent subgroups of subjects were shown in Table 2. As shown in Table 2, there was not a significant difference between the noise annoyance of subgroup a1 (10–19 years) and subgroup a2 (20–29 years), and therefore these two subgroups could be combined into one subgroup of 10–29 years (subgroup A1). Likewise, the subjects in subgroup a3 (30–39 years) and subgroup a4 (40–49 years) could be combined into one subgroup of 30–49 years (subgroup A2). The subjects in subgroup a5 (50–59 years) and subgroup a6 (60–69 years) could be combined into one subgroup of 50–69 years (subgroup A3). The subjects in subgroup c1 (primary school), subgroup c2 (junior high school), and subgroup c3 (senior high school and secondary vocational school) could be combined into one subgroup of low qualification (subgroup C1). The subjects in subgroup e2 (low income), subgroup e3 (middle income), and subgroup e4 (high income) could be combined into one subgroup with income (subgroup E2). The subjects in subgroup f2 (medium) and subgroup f3 (noisy) could be combined into one subgroup noisy (subgroup F2). The final grouping of subjects after the initial subgroups above were merged was shown in Table 1. The quantized value of each non-acoustic index in different subgroups was also shown in Table 1.

### 3.2. Correlation between Substation Noise Annoyance and Acoustic Indices

As shown in Figure 1 and Figure 2, the frequency range of substation noise went from 20 Hz to 12,000 Hz. According to the frequency range, $L_{31.5\mathrm{Hz}}$, $L_{63\mathrm{Hz}}$, $L_{125\mathrm{Hz}}$, $L_{250\mathrm{Hz}}$, $L_{500\mathrm{Hz}}$, $L_{1\mathrm{kHz}}$, $L_{2\mathrm{kHz}}$, $L_{4\mathrm{kHz}}$, $L_{8\mathrm{kHz}}$, and $L_{16\mathrm{kHz}}$ were chosen as acoustic indices in this study. The correlation between $A_{m}$ and each acoustic index was shown in Table 3. As shown in Table 3, the acoustic indices except S, T and $L_{C-A}$ were significantly linearly correlated with $A_{m}$ among the 20 acoustic indices ($R^{2}$ > 0.866). Based on these acoustic indices being significantly linearly correlated with $A_{m}$, a prediction model of annoyance from substation noise can be developed by a stepwise regression.

### 3.3. Influence of Non-Acoustic Indices on Annoyance from Substation Noise

#### 3.3.1. Age

The results of the paired samples *t*-test between the noise annoyance of different age groups were shown in Table 2. As shown in Table 2, there was a significant difference between the noise annoyance of any two groups in the three age groups (subgroups A1–A3). Noise annoyance rose with the increase in subjects’ age (see Figure 3). The research results from Gerven [20] and Miedema [1] et al. showed that noise annoyance first increased and then decreased with the increase in subjects’ age and that the inflection point was about 60 years old. The range of subjects’ age in this study was 12–62 years, so the result of this study was consistent with the research above. Subjects with different ages have different workloads and hearing losses [16]. The study from Wallenius [21] showed that subjects with higher workloads were less adaptable to noise. The study from Fields [22] and Job [9] showed that the weaker the subjects’ ability to adapt to noise, the higher their noise annoyance. The older the subjects, the more severe the hearing loss, the lower the degree of noise acceptance, and the higher the noise annoyance [20]. In this study, the proportion of students in subjects of subgroup A1 was high so the average workload of subjects in subgroup A1 was lower than that in subgroups A2 and A3. Therefore, the noise annoyance of subjects in subgroup A1 was lower than that in subgroups A2 and A3. With the increase in age, hearing loss would occur at high-frequency band and the coating effect of high-frequency noise on low-frequency noise would decrease, which could lead to an increase in noise annoyance [23]. This may be the reason that the noise annoyance of subjects in subgroup A3 was higher than that in subgroup A2.

#### 3.3.2. Gender

The result of the paired samples *t*-test between the noise annoyance of male and female groups was shown in Table 2. As shown in Table 2, there was not a significant difference between the noise annoyance of subgroup B1 (male) and subgroup B2 (female). The subjects’ gender had no significant influence on noise annoyance, which was consistent with the research results from Miedema [1], Janssen [24], and Qu [25] et al.

#### 3.3.3. Education Level

The results of the paired samples *t*-test between the noise annoyance of different education level groups were shown in Table 2. As shown in Table 2, there was a significant difference between the noise annoyance of any two groups in the three groups of subjects with different education levels (subgroups C1–C3). Noise annoyance increased with the rose of the subjects’ education level (see Figure 4), which was consistent with the research results from Miedema [1] and Sieber [26] et al. The people with a higher education level paid more attention to noise and its influence on daily life and had a higher requirement for the quality of sound environment during work and rest [27,28], so they had higher noise annoyance.

#### 3.3.4. Noise Sensitivity

The result of the paired samples *t*−test between the noise annoyance of the low noise sensitivity group and the high noise sensitivity group was shown in Table 2. As shown in Table 2, there was a significant difference between the noise annoyance of subgroup D1 (low noise sensitivity) and subgroup D2 (high noise sensitivity). Noise annoyance of subjects in the high noise sensitivity group was higher than that in the low noise sensitivity group (see Figure 5), which was consistent with the research results from Di [19], Kamp [29], Janssen [24], Benz [30], and Jakovljevic [31] et al. Noise sensitivity could be regarded as a physiological or psychological state of the subject which can increase the degree of response to noise. The higher the noise sensitivity score of subjects, the greater the degree of response to a same noise and the higher the noise annoyance of subjects [32].

#### 3.3.5. Income

The result of the paired samples *t*−test between the noise annoyance of the two groups of subjects with income and without income was shown in Table 2. As shown in Table 2, there was a significant difference between the noise annoyance of subgroup E1 (without income) and subgroup E2 (with income). The noise annoyance of subjects without income was lower than that of those with income (see Figure 6). A previous study [31] showed that the people with less stress in their daily life were not susceptible to be influenced by noise and to experience annoyance. In this study, the subjects without income were all students. Compared to subjects with income obtained by formal work, students had lower stress levels, so the annoyance of the latter was lower than that of the former at a same noise exposure.

#### 3.3.6. Noisy Degree in the Workplace

The result of the paired samples *t*−test between the noise annoyance of the quiet group and that of the noisy group was shown in Table 2. As shown in Table 2, there was a significant difference between the noise annoyance of subgroup F1 (quiet) and subgroup F2 (noisy). The noise annoyance of subjects in the quiet group was lower than that in the noisy group (see Figure 7), which was consistent with the research results from Jakovljevic [31] et al. A previous study [21] showed that the people whose workplace had a higher noisy degree would have higher annoyance at a same noise exposure.

### 3.4. Prediction Model of Annoyance from Substation Noise

The study identified the 17 acoustic indices as being significantly linearly correlated with noise annoyance and 5 non-acoustic indices (such as age, education level, noise sensitivity, income, noisy degree in the workplace) as significantly influencing noise annoyance in Section 3.2 and Section 3.3. Based on these indices, a prediction model of annoyance from substation noise was developed by a stepwise regression (see Equation (2)). As shown in Equation (2), the acoustic indices in the model were the equivalent continuous A-weighted sound pressure level ($L_{\mathrm{Aeq}}$) and the sound pressure level above 1/1 octave band of 125Hz ($L_{125\mathrm{Hz}}$). $L_{\mathrm{Aeq}}$ is a common indicator for environmental noise and can well represent the influence of substation noise [7,8]. A previous study [33] indicated that low frequency noise easily caused the annoyance of subjects. The substation noise energy mainly concentrates on 100Hz and its harmonic frequency [8], so $L_{125\mathrm{Hz}}$ can well represent the influence of low frequency substation noise. Therefore, $L_{\mathrm{Aeq}}$ and $L_{125\mathrm{Hz}}$ finally entered the prediction model
(2)$$A_{m}=0.16L_{\mathrm{Aeq}}+0.04L_{125\mathrm{Hz}}+0.32Age+0.07Edu+0.25NS+0.21Econ+0.09Envi-7.19$$
where Age, Edu, NS, Econ, Envi are the quantized values of a subject’s age, education level, noise sensitivity, income, and noisy degree in the workplace, respectively (See Table 1).

According to Equation (2), the acoustic indices in the model were $L_{\mathrm{Aeq}}$ and $L_{125\mathrm{Hz}}$. The absolute values of their standard regression coefficients were 0.76 and 0.18, respectively (see Table 4). Their influence weights on substation noise annoyance were 65% and 15%, respectively (see Table 4). The total influence weight of acoustic factors on substation noise annoyance was 80%. The non-acoustic indices in the model were age, education level, noise sensitivity, income, and noisy degree in the workplace. The absolute values of their standard regression coefficients were 0.09, 0.02, 0.05, 0.05, and 0.02, respectively (see Table 4). Their influence weights on substation noise annoyance were 8%, 2%, 4%, 4%, and 2%, respectively (see Table 4). The total influence weight of non-acoustic factors on substation noise annoyance was 20%. Vallin et al. [34] developed a prediction model of annoyance from railway noise through listening tests and the total influence weight of non-acoustic factors on annoyance was 26.1%. It can be seen that non-acoustic factors significantly influence individual noise annoyance and the influence weight of non-acoustic factors on annoyance from different noise is different.

## 4. Conclusions

To identify the non-acoustic indices significantly influencing noise annoyance, substation noise was chosen as a case study and listening tests were conducted. A prediction model of annoyance from substation noise including acoustic and non-acoustic factors was developed. Results showed that age, education level, noise sensitivity, income, and noisy degree in the workplace were the non-acoustic factors significantly influencing noise annoyance. The influence weights of acoustic and non-acoustic factors on substation noise annoyance were 80% and 20%, respectively. The influence weights of $L_{\mathrm{Aeq}}$ and $L_{125\mathrm{Hz}}$ on substation noise annoyance were 65% and 15%, respectively. The influence weights of age, education level, noise sensitivity, income, and noisy degree in the workplace on substation noise annoyance were 8%, 2%, 4%, 4%, and 2%, respectively. The result of this study can provide a basis for influencing factors identification and prediction of noise annoyance.

This study only explored the influence of age, gender, education level, noise sensitivity, income, and noisy degree in the workplace on noise annoyance. The influence of other non-acoustic factors such as visual factors, noisy degree at home, attitudes towards noise source, etc., on annoyance could be further studied. The prediction model of noise annoyance developed in this study was only applicable to substation noise. Prediction models of noise annoyance which are applicable to other types of noise could be further studied.

## Figures and Tables

**Figure 1 ijerph-19-08394-f001:**
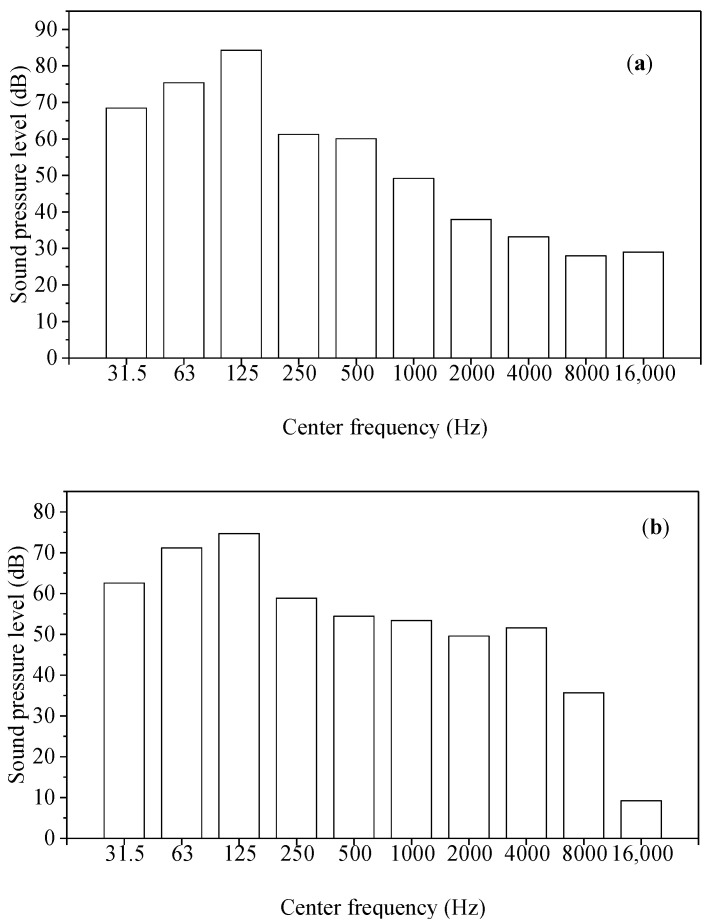
A 1/1 octave band spectra of two typical noise samples (**a**) 500 kV substation noise (**b**) 1000 kV substation noise.

**Figure 2 ijerph-19-08394-f002:**
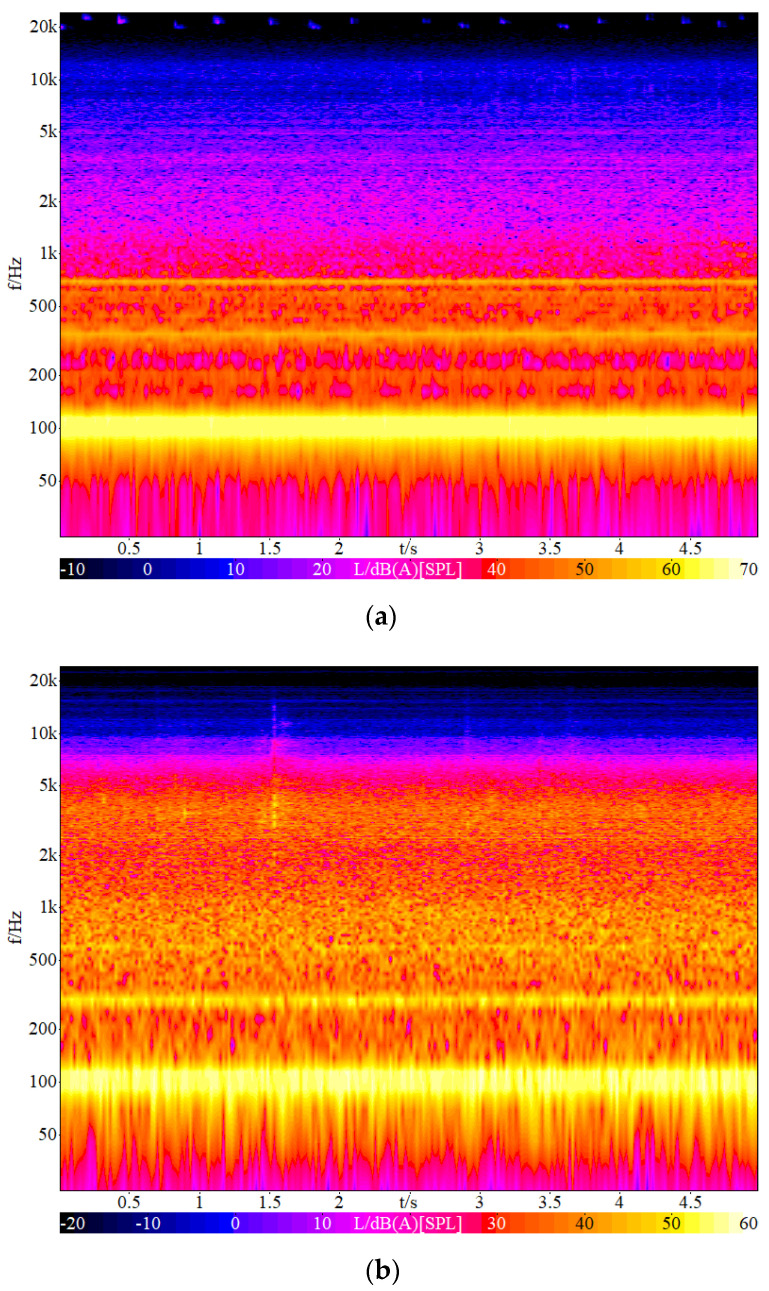
Time-frequency signal of two typical noise samples (**a**) 500 kV substation noise (**b**) 1000 kV substation noise.

**Figure 3 ijerph-19-08394-f003:**
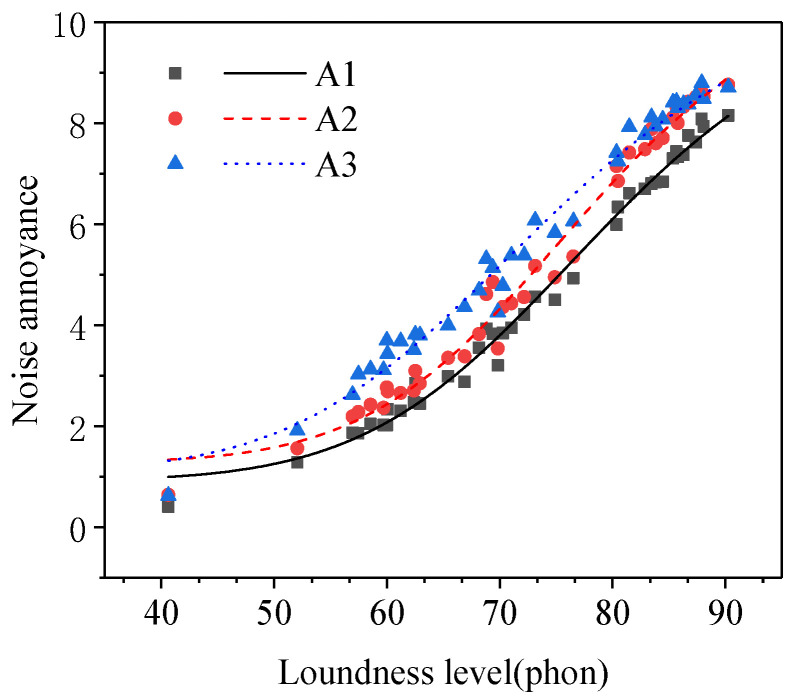
Noise annoyance of three groups of subjects with different ages (subgroups A1–A3).

**Figure 4 ijerph-19-08394-f004:**
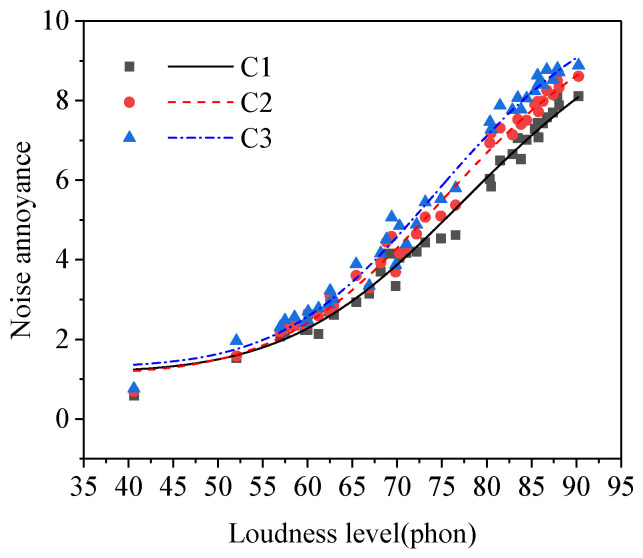
Noise annoyance of three groups of subjects with different education levels (subgroups C1–C3).

**Figure 5 ijerph-19-08394-f005:**
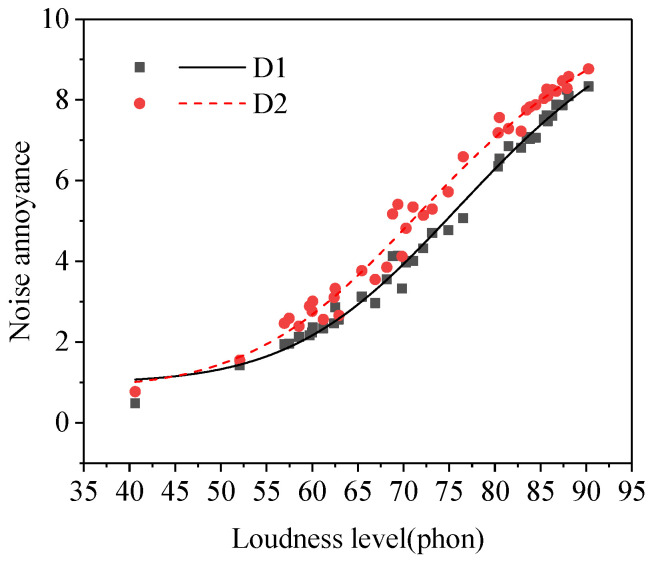
Noise annoyance of two groups of subjects with different noise sensitivities (subgroups D1–D2).

**Figure 6 ijerph-19-08394-f006:**
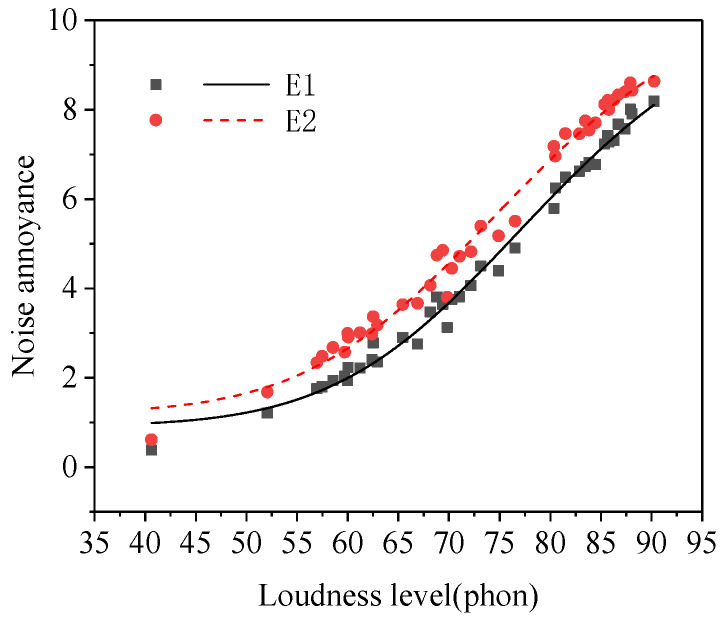
Noise annoyance of two groups of subjects with different incomes (subgroups E1–E2).

**Figure 7 ijerph-19-08394-f007:**
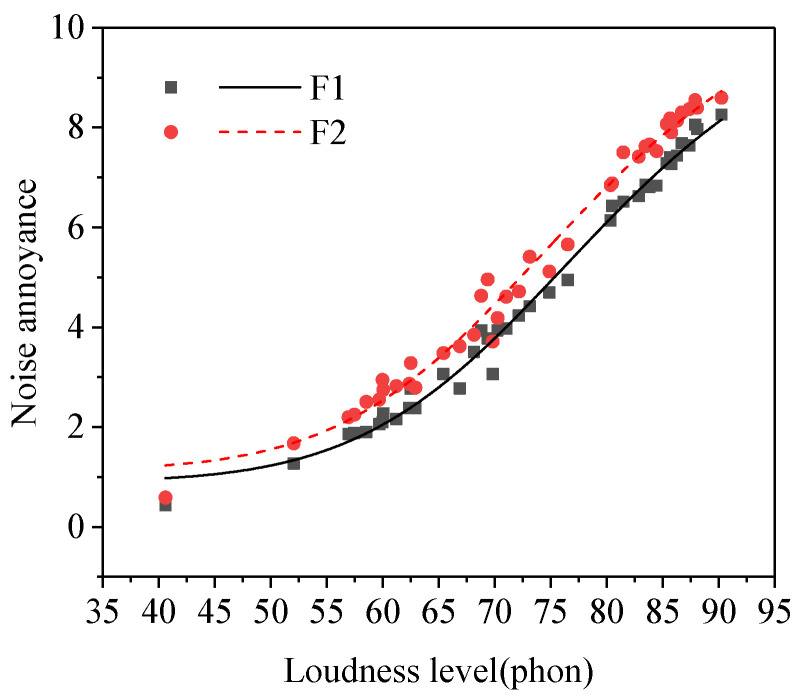
Noise annoyance of two groups of subjects with different noisy degrees in the workplace (subgroups F1–F2).

**Table 1 ijerph-19-08394-t001:** Basic information and grouping of subjects.

Non-Acoustic Indices	The Grouping Initially (The Number of Subgroups)	The Amount of Subjects (Proportion)	The Grouping after Mergence (The Number of Subgroups)	The Quantized Value ^1^
Age	10–19 Years (a1)	34 (13.8%)	10–29 Years (A1)	1
	20–29 Years (a2)	63 (25.6%)		
	30–39 Years (a3)	44 (17.9%)	30–49 Years (A2)	2
	40–49 Years (a4)	43 (17.5%)		
	50–59 Years (a5)	32 (13.0%)	50–69 Years (A3)	3
	60–69 Years (a6)	30 (12.2%)		
Gender ^2^	Male (b1)	101 (41.1%)	Male (B1)	1
	Female (b)	145 (58.9%)	Female (B2)	2
Education level	Primary school (c1)	40 (16.3%)	Low qualification (C1)	1
	Junior high school (c2)	69 (28.0%)		
	Senior high school and secondary vocational school (c3)	41 (16.7%)		
	Undergraduate and junior college students (c4)	40 (16.3%)	Undergraduate and junior college students (C2)	2
	Graduate (c5)	56 (22.7%)	Graduate (C3)	3
Noise sensitivity ^3^	Low noise sensitivity (d1)	109 (87.9%)	Low noise sensitivity (D1)	1
	High noise sensitivity (d2)	15 (12.1%)	High noise sensitivity (D2)	2
Income	Without income (e1)	74 (30.1%)	Without income (E1)	1
	Low income (e2)	101 (41.0%)	With income (E2)	2
	Medium income (e3)	41 (16.7%)		
	High income (e4)	30 (12.2%)		
Noisy degree in workplace	Quiet (f1)	108 (43.9%)	Quiet (F1)	1
	Medium (f2)	46 (18.7%)	Noisy (F2)	2
	Noisy (f3)	92 (37.4%)		

^1^ Each non-acoustic index was assigned with an equal interval in different subgroups. The quantized value of a single non-acoustic index did not change the standard regression coefficient of each non-acoustic index in the prediction model of noise annoyance. ^2^ Two subgroups of gender were not combined in this study. ^3^ According to the score of subjects in the Weinstein noise sensitivity scale, 110 was regarded as the threshold distinguishing high and low noise sensitivity of subjects [19].

**Table 2 ijerph-19-08394-t002:** The results of paired samples *t*-test of noise annoyance between different groups.

Paired Groups	*t* Value	*p* Value	Paired Groups	*t* Value	*p* Value
a1 vs. a2	0.84	0.35	e1 vs. e2	−8.5	<0.05
a2 vs. a3	−13.8	<0.05	e2 vs. e3	1.7	0.09
a3 vs. a4	−0.26	0.88	e3 vs. e4	1.10	0.24
a4 vs. a5	−4.9	<0.05	f1 vs. f2	−12.7	<0.05
a5 vs. a6	−0.76	0.45	f2 vs. f3	−1.23	0.21
b1 vs. b2	1.6	0.12	A1 vs. A2	−14.3	<0.05
c1 vs. c2	−0.1	0.92	A2 vs. A3	−9.6	<0.05
c2 vs. c3	0	0.96	C1 vs. C2	−6.1	<0.05
c3 vs. c4	−5.8	<0.05	C2 vs. C3	−10.4	<0.05
a3 vs. a4	−0.26	0.88	E1 vs. E2	−22.7	<0.05
c4 vs. c5	−10.4	<0.05	F1 vs. F2	−17.4	<0.05
d1 vs. d2	−12.3	<0.05			

**Table 3 ijerph-19-08394-t003:** The correlation between substation noise annoyance and each acoustic index.

Acoustic Index	Determination Coefficient $\left(R^{2}\right)$	*p* Value
N	0.870	<0.05
F	0.921	<0.05
*R*	0.919	<0.05
S	0.052	0.09
T	0.060	0.07
SPL	0.931	<0.05
$L_{\mathrm{Aeq}}$	0.959	<0.05
$L_{\mathrm{Ceq}}$	0.933	<0.05
$L_{C-A}$	0.007	0.27
$L_{\max}$	0.908	<0.05
$L_{31.5\mathrm{Hz}}$	0.945	<0.05
$L_{63\mathrm{Hz}}$	0.884	<0.05
$L_{125\mathrm{Hz}}$	0.875	<0.05
$L_{250\mathrm{Hz}}$	0.935	<0.05
$L_{500\mathrm{Hz}}$	0.937	<0.05
$L_{1\mathrm{kHz}}$	0.939	<0.05
$L_{2\mathrm{kHz}}$	0.961	<0.05
$L_{4\mathrm{kHz}}$	0.958	<0.05
$L_{8\mathrm{kHz}}$	0.950	<0.05
$L_{16\mathrm{kHz}}$	0.866	<0.05

**Table 4 ijerph-19-08394-t004:** The standard regression coefficient of each variable in the model.

Variable	Standard Regression Coefficient	Influence Weight
$L_{\mathrm{Aeq}}$	0.76	65%
$L_{125\mathrm{Hz}}$	0.18	15%
Age	0.09	8%
Edu	0.02	2%
NS	0.05	4%
Econ	0.05	4%
Envi	0.02	2%

## Data Availability

The data that supported the findings are available from the corresponding author upon reasonable request.

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
