# Peer review of "Influencing Factors Identification and Prediction of Noise Annoyance—A Case Study on Substation Noise"

_ijerph, 2022, doi:10.3390/ijerph19148394_

Round 1
Reviewer 1 Report
This study examines the acoustic and non-acoustic factors that affect annoyance, but several significant problems exist.
The reasons are as follows, and I propose to reject this study because I think it needs to be fundamentally corrected.
- The acoustic characteristics of substation noise are not shown at all. The results discuss the importance of the sound pressure level at 125 Hz, but there is no data to support this discussion from the viewpoint of acoustic characteristics. Also, the sound pressure levels of the stimuli used in the experiments are not given. Information is needed on what range of sound pressure levels was presented. Furthermore, as shown in Table 3, although there is a very high correlation between octave band level and annoyance over a very wide frequency range, the substation noise used in the experiment has such a wide range of frequency components?
- The first part of section 2.5 describes the rules for excluding data and/or subjects, but it is unclear how many responses were removed and how many subjects were removed. Since the analysis is based on the data after removing the respondents, we do not know the number of data used in the analysis, which is what the reader wants to know.
- The t-test results shown in Table 2 are used to categorize the participants, but I think the rule is ambiguous. For example, there is no significant difference between a1 and a3, but why were they classified into different groups?
- Section 3.3.1 discusses the effect of hearing loss. Noise-induced hearing loss affects approximately 2 kHz, while age-related hearing loss affects higher frequencies. Are such high-frequency components predominant in the substation sound? I think an examination corresponding to the physical characteristics of substation sound is needed.
- When dealing with an extremely large number of variables, as in this study, a stepwise method that reduces the number of variables should not be used.
Reviewer 2 Report
Describe better how you recorded the noise.
Why didn't you enter the time history of the signal you measured and the frequency trend?
Please describe the measured sound levels better.
Describe better how you performed the listening tests, how you calibrated the signal that is heard by the people who perform the test, this procedure is very important to understand how you performed the listening tests.
Rewriting the paragraph (3 Results and discussion) should be split in two and highlight what exactly was done.
Describe what your contribution to research is.
Reviewer 3 Report
Ref. Review: IJERPH-1745271
Paper Title: Influencing factors identification and prediction of noise annoyance – a case study on substation noise
Dear Authors and Editor,
Based on the text exposed in the entitled paper: " Influencing factors identification and prediction of noise annoyance – a case study on substation noise", I recommend major revisions before acceptance and publishing in Environmental Research and Public Health.
The work presented in this paper investigated how subjects regarding noise annoyance rated the substation noise through listening tests. Non-acoustic indices were divided into groups and helped to characterise the substation noise annoyance.
This work aims to verify how socio-economic information influences subjects’ perception responses according to a specific type of sound source that generates annoyance. This study validates how the population perceives different acoustic characteristics combined with non-acoustic information in urban scenarios.
Please consider the following recommendations for the improvement of this manuscript:
Overall text:
1) Please leave the reference number in normal letter style, not superscript. Sometimes if hard to read and find the references in your text.
Introduction:
2) Could you please do an overview of studies that investigated noise annoyance using:
· Non-acoustic parameters,
· Socio-economic parameters,
· Type of sound source that generates noise annoyance.
Please add and connect these references with the discussion section, supporting your findings.
Materials and methods:
3) 2.1 Subject recruitment and information collection: please inform me how you recruit the participants? Who are the participants? Did they get any reward for participating in your listening experiment?
4) 2.3 Room and equipment for listening tests: referring to the sentence “Tests showed that the playback system had no significant distortion from 20 Hz to 20,000 Hz and would not affect the reproduction of low-frequency components of noise samples”. How did you test the measurement/reproduction chain? Please add references.
5) 2.4 Experimental procedures: the listening experiment used sounds until 90 phons, did the procedure pass through an ethics committee? Could you please explain what the content of the consent document was? Please add this information to the text.
6) 2.5 Statistical analysis:
· Regarding the misjudgment of noise samples, did the participants realise audiometry before the experiment to ensure the rating of normal hearing subjects?
· Regarding the sentence “subject acted as a ‘measuring instrument’ for noise annoyance”, thinking of the subject as a measuring instrument. You should also observe the physiological aspects of the subject. Knowing that each subject has a different auditory system, regarding sizes and so on, can directly influence the Head-Related Transfer Function (HRTF), which significantly affects how loud and the directivity of perceived sound sources. In this case, it is indicated to measure the HpTF to ensure that all participants hear sound signals with the same loudness. This procedure calibrates the loudness of the reproduced stimuli. How did the authors confirm that the loudness of the reproduced stimuli was “the same” for each participant? If the authors did not follow any calibration procedure, I highly recommend clarifying this point in the text and reforming the sentence about the subject at the measurement instrument. Please also inform that the perception of sound sources also depends on the hearing system (physiological aspect), but this study refers only to the psychological element; please add references about physiological and psychological elements.
Masiero B. S. Individualized Binaural Technology. PhD Thesis, vol.13, RWTH Aachen University, Aachen, 2012 Available at: https://index.ub.rwth-aachen.de/TouchPointClient_touchpoint/singleHit.do?methodToCall=showHit&curPos=1&identifier=2_SOLR_SERVER_325708094
Masiero B. S., Fels J. Perceptually Robust Headphone Equalization for Binaural Reproduction. In Convention of the Audio Engineering Society 130, 2011. Available at https://www.aes.org/e-lib/online/browse.cfm?elib=15855
· After data collection, there was a step of data treatment before statistical analysis?
· Please inform all the information mentioned above in the text and add references.
Results and Discussion:
7) Table 1: If only 138 subjects (informed in section 2.1) were willing to participate in the experiment, why are you informing the result of 246 subjects? Age range 10-19 Years (a1): children and adolescents also participated in the survey; there was a special consent for these subjects?
8 8) Table 2: could you also inform the t-test value? Here you are showing just if the p-value is significant. But you are not telling how the acoustic condition regarding the differences was? Did you check all samples together? Did to divide into sound level groups? Please be more specific.
9 9) Table 3: once again, is it about the overall condition or regarding sound ranges? Please be evident in the text.
Conclusion:
10) What are the limitations of this study?
Reviewer 4 Report
Dear authors,
thank you for sending this paper through. The topic is very interesting, although some points shall be improved. It would be worth it if the results are compared to any reference standard and/or criteria defined by previous literature. This is valid for all the graphs. The conclusions shall be extended and try to outline how this research can be improved, what are the points that are not deeply explored and other researchers can take over.
Round 2
Reviewer 1 Report
In my last review, I could not judge this manuscript's value because there was no acoustic information on the substation noises. However, the modified version shows those, and my doubt and confusion about the paper have been clarified.
Reviewer 2 Report
Dear author, thank you for replying to my comments.
But in rereading the paper I wrote these observations:
Check dB or dBA in the text
check editing
check english.
Fig. 2 the numbers are too small
Background noise 25 dBA. How did you measure it?
(type of sound level meter)
ISO 15666 can you insert the reference?
Can you describe Artemis software how it works?
Page 12, ??? , ??? , ?? , ???? , ???, you can better explain how to put them in the form at what they are worth. Could you insert a summary table?
You can better explain how you performed the listening test, how you played the sample sound; with headphone or speaker?
How did you equalize the sample sound?
Could you describe this procedure better? Because the procedure provides an indication if the noise is annoying or not.
Could you post a picture of how you perform the test?
Thanks for your patience.
Reviewer 3 Report
Ref. Review: IJERPH-1745271
Paper Title: Influencing factors identification and prediction of noise annoyance – a case study on substation noise
Dear Authors and Editor,
Based on the text exposed in the entitled paper: " Influencing factors identification and prediction of noise annoyance – a case study on substation noise", I recommend major revisions before acceptance and publishing in Environmental Research and Public Health.
The work presented in this paper investigated how subjects regarding noise annoyance rated the substation noise through listening tests. Non-acoustic indices were divided into groups and helped to characterise the substation noise annoyance.
This work aims to verify how socio-economic information influences subjects’ perception responses according to a specific type of sound source that generates annoyance. This study validates how the population perceives different acoustic characteristics combined with non-acoustic information in urban scenarios.
Please consider the following recommendations for the improvement of this manuscript:
Overall text:
1) Please leave the reference number in normal letter style, not superscript. Sometimes if hard to read and find the references in your text.
The IJERPH has a guideline on how the format of the reference in the text should be. Please follow the suggested configuration of the journal.
https://www.mdpi.com/journal/ijerph/instructions#referees
https://susy.mdpi.com/user/manuscripts/upload/5ebb983bd6c156eb81113fcf5965c5d6?&form%5Bjournal_id%5D=6 (Microsoft word template and LaTex template)
Materials and methods:
2) 2.2 Noise samples: Figure 2 - Could you please highlight the frequency differences between figures a and b? As far as I can see there is evidence at 100 Hz frequency, also in figure b, there is an event at 1.5, 3, 3.5, and 3.7 seconds that are forming sound signatures at 10k. Could you explain these characteristics in the text?
3) 2.3 Room and equipment for listening tests: referring to the sentence “Tests showed that the playback system had no significant distortion from 20 Hz to 20,000 Hz and would not affect the reproduction of low-frequency components of noise samples”. How did you test the measurement/reproduction chain? Please add references.
The authors answer in the text: “The equipment used in this study would pass a rigorous test by the instrument supplier before they were delivered”.
Reviewer answer: This is expected from all equipment received from the suppliers. But you must be sure through system calibration if you are reproducing noise/sound levels that really correspond to reality. Please check the sound level accuracy on listening tests. This can be done by calibration on the chain of measurement/reproduction.
Please check the following publications:
· Dietrich P. Uncertainties in acoustical transfer functions: Modeling, measurement and derivation of parameters for airborne and structure-borne sound. D82 Diss. RWTH Aachen University. Logos Verlag Berlin GmbH, 2013. ISBN 978-3-8325-3551-3.
· Institute of Technical Acoustics. ITA-Toolbox-An Open-Source MATLAB Toolbox for Acousticians, RWTH-Aachen University, 2008.
The authors indicated a reference to the following text that is not mentioning the content of the reported text:
“The results of the test showed that the playback system did not have a distinct distortion from 20 Hz to 20,000 Hz and would not affect the reproduction of low-frequency components of noise samples[15].”
Reference 15 is about a guideline regarding listening tests. In this guideline, there are no indications of the frequency range or frequencies for the reproduction of low frequencies.
4) 2.5 Statistical analysis: “If the difference between any two evaluation results for a same noise sample from a same subject is greater than 2, the noise sample would be regarded as a misjudged noise sample from this subject and the data of misjudged noise samples will be removed in data analysis. If the ratio of the amount of misjudged noise samples to the total amount of noise samples is higher than 30%, this subject would be regarded as an invalid subject and all data from the subject would be eliminated.” Please add references for these sentences.
Results and Discussion:
5) Table 1: If only 138 subjects (informed in section 2.1) were willing to participate in the experiment, why are you informing the result of 246 subjects? Age range 10-19 Years (a1): children and adolescents also participated in the survey; there was a special consent for these subjects? Please check: https://www.mdpi.com/journal/ijerph/instructions#referees
6 6) 3.2 Correlation between substation noise annoyance and acoustic indices: Please connect Table 2 to the spectrograms shown in Figures 2a and 2b
